# Pearls and Pitfalls of Adaptive Optics Ophthalmoscopy in Inherited Retinal Diseases

**DOI:** 10.3390/diagnostics13142413

**Published:** 2023-07-19

**Authors:** Helia Ashourizadeh, Maryam Fakhri, Kiana Hassanpour, Ali Masoudi, Sattar Jalali, Danial Roshandel, Fred K. Chen

**Affiliations:** 1Department of Ophthalmology, Mayo Clinic, Rochester, MN 55905, USA; ashourizadeh.helia@mayo.edu; 2Ophthalmic Research Center, Research Institute for Ophthalmology and Vision Sciences, Shahid Beheshti University of Medical Sciences, Tehran 16666, Iran; maryam.fakhri88@gmail.com (M.F.); kiana.hassanpour@gmail.com (K.H.); 3Stein Eye Institute, David Geffen School of Medicine, University of California, Los Angeles, CA 90095, USA; masoudi.a@umsu.ac.ir; 4Department of Physics, Central Tehran Branch, Islamic Azad University, Tehran 19558, Iran; sahab.jalali@gmail.com; 5Centre for Ophthalmology and Visual Science, The University of Western Australia, Nedlands, WA 6009, Australia; danial.roshandel@lei.org.au; 6Ocular Tissue Engineering Laboratory, Lions Eye Institute, Nedlands, WA 6009, Australia; 7Centre for Eye Research Australia, Royal Victorian Eye and Ear Hospital, Melbourne, VIC 3002, Australia; 8Ophthalmology, Department of Surgery, University of Melbourne, Melbourne, VIC 3010, Australia

**Keywords:** adaptive optics, photoreceptor, retinal dystrophy, inherited retinal disease, rod-cone dystrophy

## Abstract

Adaptive optics (AO) retinal imaging enables individual photoreceptors to be visualized in the clinical setting. AO imaging can be a powerful clinical tool for detecting photoreceptor degeneration at a cellular level that might be overlooked through conventional structural assessments, such as spectral-domain optical coherence tomography (SD-OCT). Therefore, AO imaging has gained significant interest in the study of photoreceptor degeneration, one of the most common causes of inherited blindness. Growing evidence supports that AO imaging may be useful for diagnosing early-stage retinal dystrophy before it becomes apparent on fundus examination or conventional retinal imaging. In addition, serial AO imaging may detect structural disease progression in early-stage disease over a shorter period compared to SD-OCT. Although AO imaging is gaining popularity as a structural endpoint in clinical trials, the results should be interpreted with caution due to several pitfalls, including the lack of standardized imaging and image analysis protocols, frequent ocular comorbidities that affect image quality, and significant interindividual variation of normal values. Herein, we summarize the current state-of-the-art AO imaging and review its potential applications, limitations, and pitfalls in patients with inherited retinal diseases.

## 1. Introduction

Incorporation of the adaptive optics (AO) technology into ophthalmic devices has led to the development of non-invasive ultra-high-resolution imaging of retinal cells and blood vessels with minimal image degradation by higher-order monochromatic aberrations. AO was first incorporated into ophthalmoscopy by Liang et al. in 1997 [1], and since then, it has gained popularity as a structural endpoint in clinical trials (www.ClinicalTrials.gov; NCT03349242, NCT00254605, and NCT01866371; accessed on 15 December 2022). Notable knowledge advances have been made by using AO in the early detection of retinal abnormalities. Numerous studies demonstrated that AO could detect early changes in photoreceptor density before visible clinical signs or abnormalities in conventional imaging techniques [2,3]. Collecting indirectly scattered light from adjacent cells with different refractive indices, AO can also provide valuable information on retinal vasculature. High-contrast perfusion maps provide information on retinal blood vessels’ structure, including lumen width, wall thickness and wall-to-lumen ratio, branching, tortuosity, microaneurysms, as well as function, such as resting and stimulus-evoked changes in blood flow (neurovascular coupling) [4,5,6]. Additionally, the unique capabilities of AO optical coherence tomography angiography (OCTA) in quantitative imaging of choriocapillaris has been demonstrated in inherited retinal diseases (IRDs) [7].

IRDs comprises a wide range of retinal degenerations, which collectively is the leading cause of visual disability and legal blindness in the working-age population [8]. Currently, one gene-based therapy has been approved for the treatment of *RPE65*-associated retinopathy [9,10] while gene-based therapies for other IRD genes are still under investigation [11,12]. AO retinal imaging can be useful for tracking a disease’s natural history at the cellular level and defining outcome measures for therapeutic interventions in clinical trials [13]. Despite the lack of a standardized imaging protocol and image analysis methods in AO, the detailed visualization of retinal cell survival provided by AO offers potential applications in measuring efficacy in stem cell and gene-targeted therapies. This comprehensive review discusses the main concepts of AO in ophthalmic imaging, its findings in IRDs, and the pitfalls and limitations of AO in research and clinical practice.

## 2. Basic Concepts

### 2.1. Principles of Adaptive Optics

As first conceptualized by astronomer Horace Babcock in the 1920s, atmospheric media cause turbulences in the light waves originating from stars, which leads to low image resolution in ground-based telescopes [14]. Similarly, distortions and irregularities on the ocular surface and within the eye make it an imperfect optical system. There are three key components in every AO system, which detect aberrations in real time and adapt to enhance the image quality. A wavefront sensor detects the optical aberrations, and a wavefront corrector compensates for the aberrations. The controlling system coordinates the sensor and controller to provide the best image quality with low aberrations. In sensor-based AO, also known as hardware AO, wavefront sensors are used to measure the aberrations of the reflected light after passing through the ocular media and cornea. The calculated aberration is transferred to the controller and translates into signals to be sent to a deformable mirror and acts as a guide to ongoing corrective changes of the mirror. Then, the modified wavefront is detected by the camera and a high-resolution image is captured (Figure 1).

In sensorless AO and computational AO systems, aberrations are detected based on optimization algorithms, and the distortion degree of the wavefront is estimated according to the properties of multiple serially taken consecutive images. Sensorless AO has a major drawback in terms of speed since it necessitates a series of images to gauge and adjust a specific set of aberration modes. Sensor-based AO systems are more optically complex and costly devices with almost 10 times higher speeds compared to sensorless AO [15]. However, the retinal images captured through sensorless adaptive optics are comparable to those obtained with wavefront sensor-based control [16].

### 2.2. Technologies and Devices

#### 2.2.1. AO Flood Illumination Ophthalmoscopy (AOFIO)

The only commercially available AO fundus camera (rtx1, Imagine Eyes, Orsay, France) illuminates a small region of the retina and captures 40 images from a 4° × 4° field over 2 or 4 s (19 or 9.5 frames per second, respectively). Since each image is acquired in milliseconds, fine ocular movements do not affect image quality. Artifact-free images are aligned and blended using built-in software to increase the signal-to-noise ratio (SNR). The processed image is a magnified high-resolution image of cone photoreceptors or retinal vessels (Figure 2) [17]. This device gives an enface image of a 4° × 4° field of retina, corresponding to a 1.2 × 1.2 mm square. Each acquisition lasts 4 s (2 s in the upgraded version) during which 40 live images are taken. Using the cross-correlation method and automated reference frame selection by the image processing software provided by the device’s manufacturer (AOdetect), in every series of raw image sequences, poor-quality images are omitted, and frames with acceptable resolution and the least distortion are averaged to give a final image with the highest amount of signal-to-noise ratio [18,19].

Images from adjacent fields may be captured and montaged to create a map of the cone mosaic or retinal vasculature. The major drawback of AOFIO is that light scattered from adjacent structures may reduce the contrast of the image (axial resolution is about 300 µm). In addition, although the theoretical resolution of AOFIO is 1.5 µm, the actual resolution is not high enough to visualize small cones with the highest packing density located within the central 2° of the fovea, as well as rod photoreceptors, which have a diameter of 1–2 µm. In addition, measurements might be affected by a large number of intervening rods beyond 6° from the fovea. Hence, although cones might be visualized at closer than 200 µm from the fovea (within 1°) and rods might be faintly visible in ideal conditions [20], the usefulness of AOFIO is generally limited to eccentricities between 2° and 5° [21].

#### 2.2.2. AO Scanning Laser Ophthalmoscopy (AOSLO)

In contrast to AOFIO, AOSLO devices are not yet commercially available. In AOSLO, small regions (approximately 1.5° × 1.5°) of the retina are illuminated, and backscattered lights are detected at the same plane as the illumination. Hence, AOSLO produces images with higher contrast compared to AOFIO and permits visualization of rods and foveal cones with a transverse and axial resolution of about 2.5 µm and 100 µm, respectively. However, the small scanning field requires more scans to obtain cone mosaics, which requires the time-consuming task of creating montages to allow identification of landmarks for image registration [22]. While both AOFIO and confocal AOSLO (cAOSLO) rely on the reflectivity of the photoreceptors (which depends on their intact structure and waveguide properties) for visualization, non-confocal split-detector AOSLO (SD-AOSLO) shows the cone inner segment ellipsoid independent of reflectivity/waveguiding. Hence, SD-AOSLO can show cones without the time and morphology-dependent reflectivity changes. However, SD-AOSLO is unable to resolve rods [23]. Table 1 provides major differences between AOFIO and AOSLO.

#### 2.2.3. AO Optical Coherence Tomography (AO-OCT)

AO-OCT provides a five-times higher lateral resolution in comparison to conventional SD-OCT [24,25], and a better axial resolution than AOSLO [26], which confers a distinct benefit compared to conventional OCT as well as SLO and FIO. It also has an increased signal-to-noise ratio, smaller speckle size, and increased sensitivity to weak reflections, all of which enable more precise and accurate detection and visualization of microscopic structures in the retina [27]. Image quality is limited due to high rates of motion artifacts from high magnification and during the time-consuming image acquisition. By recording *en face* OCT and SLO images simultaneously and correcting transverse motion with SLO images, higher lateral and axial resolution with minor residual motion artifacts can be achieved [28]. Using a Wavefront sensorless AO-OCT on non-mydriatic pupils, the photoreceptor mosaic at eccentricities as small as 1° could be visualized [29].

## 3. Acquisition, Analysis, and Interpretation

### 3.1. Image Acquisition Protocols

Multiple factors, such as focus plane, imaging pattern, and imaging area, should be considered when obtaining the highest-quality image at the desired region (s) of interest. These factors will be further elaborated in the next sections.

#### 3.1.1. Photoreceptor

The level of focus for photoreceptor imaging ranges between 40 and 100 µm proximal to the Bruch’s membrane. The choice of imaged area depends on the purpose of the imaging. Localized imaging can be used to measure cone parameters at a specific predefined location (s) for structure–function correlation or natural history studies. Alternatively, wider areas can be imaged to produce topographic cone mosaic maps (Figure 3). The test duration and scan area should be balanced to reduce patient discomfort and fatigue. Although AOFIO captures relatively larger retinal areas compared to AOSLO, the image quality may degrade toward the edges of the image frame. Hence, many researchers use overlapping image tiles to overcome this potential limitation. Different imaging protocols have been used by rtx1 users. The operator adjusts the internal fixation target to shift a patient’s gaze systematically through a series of coordinates to allow methodical acquisition of images at various retinal loci of interest. For visualizing photoreceptors in larger areas, multiple AO images with overlapping regions are required to create a wide-field AO montage. A cluster of consecutive images with 0.5° to 2° of overlap are used to create the wide-field AO montage. Different imaging patterns may be used depending on the purpose of the imaging (Figure 3). For example, imaging of the central 3° (Figure 3A) is useful for detecting photoreceptor changes in the fovea in early-stage maculopathy, while extended fields (Figure 3B–D) are required in peripheral photoreceptor diseases, such as retinitis pigmentosa (RP). The time required for this process is even greater for AOSLO systems where the field of view can be smaller than 1° × 1° in each single image. As in AOFIO, AOSLO devices capture multiple images from the same retinal area at the same time. Each image set is called a “tile”, and usually 50–150 tiles are acquired in a single session of imaging [30]. Montaging can be performed either manually by expert graders or automatically.

Manual montaging is a time-consuming process through which a human expert aligns overlapped tiles to each other via photo editing software packages to produce the final wide-field montage image. Then, the region of interest (ROI) is selected on the montage image for which cone metrics will then be extracted. Automatic montaging is a feature-based image stitching that predicts which two images should be aligned together according to detected key points and descriptors. The process can take as long as minutes to hours based on image qualities and the applied algorithm. Many devices rely on scale invariant feature transform (SIFT) [31], but newer oriented rotated fast brief (ORB) features has been reported to be much faster while maintaining accuracy and the final quality of montage image [30].

#### 3.1.2. Blood Vessel

Blood vessel imaging can be more challenging as there is no standard method for evaluation of the retinal vessels. Like photoreceptor imaging, an AOFIO camera can be used for blood vessel imaging using a single image frame or multiple overlapping images. The focus plane should be set at approximately 250 µm proximal to the Bruch’s membrane, which is equivalent to the retinal nerve fiber layer. Factors, such as distance from the optic nerve margin and the number of bifurcations, should be taken into account when selecting the region of interest.

Findings on retinal vasculature can be a potential marker for systemic vascular health. However, retinal vessel imaging due to a multi-plane arrangement of vessels can be challenging with conventional imaging modalities. Fluorescein angiography as the clinical gold standard for retinal vascular diseases is invasive and possesses some limitations. By contrast, vascular imaging using adaptive optics imaging as a non-invasive method can be challenging due to the lack of standard metrics and protocols. Without using any exogenous contrast agents, measuring the parafoveal capillary leukocyte velocity by non-confocal AOSLO and the use of motion contrast to create non-invasive parafoveal perfusion maps provides a high-resolution image of the retinal capillary network [32,33]. Tam et al. evaluated the parafoveal capillary network by capturing videos at 512 × 512 pixels in the field-of-view range of 1.2° to 2.5° for 5 to 40 s [33]. Limitations of this method would be the motion artifacts and difficulty in detecting the slow perfusion vessels, which are in fact of great importance in the clinical practice [34].

### 3.2. Image Analysis

#### 3.2.1. Photoreceptor

Analysis of the photoreceptor mosaic starts with manual or automated detection and labeling of the cells. Though widely accepted, manual labeling is time-consuming and difficult to use for a large set of data, especially in the presence of diminished image quality or a pathology [35,36]. Semi-automated techniques require manual correction by an experienced observer to achieve optimum results. One of the most commonly used methods is detecting the regional maxima (pixels with the highest values in their neighborhood) of the photoreceptors’ centers followed by generating Voronoi tessellation by setting edges that connect these points using a MATLAB-based image processing technique [37]. Another method is the use of the circle Hough transform for the detection of circular structures [38]. To date, automated counting techniques have undergone significant modifications, and fully automatic or deep learning algorithms are available to detect and segment photoreceptors in both confocal and non-confocal AO imaging [39,40,41]. After all the steps of image registration, processing, or montaging are passed, images are sent to the analyzing software for automatic photoreceptor detection and calculation of the mean, maximum, and minimum of numeric values as well as percentage values and standard deviation of corresponding metrics in a defined area. In the rtx1 camera, two software packages are provided by the manufacturers, which are AODetect for photoreceptor analysis and AODetect Artery for the retinal vasculature analysis (Imagine Eyes; Orsay, Paris, France). The ROI is selected and defined by the operator by manually moving a frame with an adjustable size on the final high-resolution image. Relative to the location of the peak foveal cone density, which is defined on the montage image, the selected size of each frame for ROI may vary depending on its eccentricity. The eccentricity of each region is calculated as the relative distance between the center of that frame and the foveal center (which is considered as the reference point with fixation coordinates of *x* = 0°, *y* = 0°) and is expressed in degrees or micrometers [42]. Using algorithms based on segmentation and Delaunay triangulation, the location, size, and shape of cones are automatically detected, and a color-coded mosaic heat map can be produced in which hotter colors represent higher cone densities [43]. A cone-packing arrangement is analyzed using Voronoi diagrams [37]. Each Voronoi cell can be coded by a specific color according to the number of its neighbors, giving color-coded Voronoi domains [42]. Cones located at the edges of the selected frame, Voronoi cells of which are not fully within the ROI, are not considered for metric calculations (Figure 4).

The intensity of photoreceptor signals is also important while interpreting an AO retinal image. There is still debate about the exact origin of these bright signals, which lead to various waveguiding properties and the reflection of cones in a mosaic map. It has been stated that conditions affecting outer segment integrity and the contact of photoreceptors to the RPE apical process diminish intensity of backscattered light from the photoreceptor–RPE interface and result in lower reflectivity of cones [44,45]. Further factors, which have been accounted for in the cone-reflectivity variation in AOSLO images, are different segments length and diameter as well as coherence of the light source [46,47].

#### 3.2.2. Blood Vessel

Internal and external boundaries of blood vessel walls may be marked manually or using automated software (Figure 5). AODetect (artery mode) provides automatic wall segmentation and thickness computation with a manual adjustment. Once a point on a vessel is selected manually, the system will detect the rest of the object by itself. Mehta et al., using AODetect Artery, calculated the intra-observer variability for the wall-to-lumen ratio (WLR) and the cross-sectional area of the vascular wall (WCSA) and reported an excellent consistency [44]. Segmentation of the foveal avascular zone (FAZ) using a semi-automated procedure to extract the vessels centerline has been performed semi-automatically by Tam et al. based on the Frangi vesselness measure [33,45] Several deep learning models for automated vessel segmentation have also been introduced [46,47].

### 3.3. Outcome Measures

#### 3.3.1. Photoreceptor

Correlated metrics of the cone mosaic and their definitions are described in the following:

*Cone density* is the number of defined Voronoi cells divided by the total area of the Voronoi cells within that specific region.

*Percentage of hexagonality* is the ratio of the number of Voronoi cells with six sides to the total number of Voronoi cells within that specific region. 

*Nearest neighbor distance (NND) and farthest neighbor distance (FND)* are the distance between the center of a single cone and its closest/farthest neighbor in the adjacent Voronoi cell. These metrics are reported for an ROI as the average of NDD/FND of all Voronoi cells within that specific region.

*Intercell distance (ICD)/cone spacing* is the average distance between a single cone (the center of a Voronoi cell) and each of its neighbors (the center of adjacent Voronoi cell). The reported ICD for an ROI is the average ICD for all the Voronoi cells in that specific region.

*Number of neighbors regularity (NoNR)* is the mean number of sides of all Voronoi cells in an ROI divided by the SD of the number of sides of all Voronoi cells in that specific region.

*Voronoi cell area regularity (VCAR)* is the mean area of the bound Voronoi cells in an ROI divided by the SD of the area of the Voronoi cells in that specific region.

Cone density and spacing are the most often used parameters as outcome measures in different retinal pathologies. The reliability and variability of the measurements vary with the device, location, and cone detection method. Using the commercial device (rtx1) and software (AODetect) and manual adjustment by two independent graders, the intrasession repeatability and intersession reproducibility of all cone parameters (including cone density) were reported excellent in a normal population, with the inter-operator and intergrader agreement greater than 95% for cone density and spacing [19]. Manual correction of AODetect labeling has increased cone density by 6% [48].

Using a custom-built AOSLO device and cone detection software, the inter-observer intraclass correlation coefficients (ICCs) for cone density and hexagonal Voronoi domain were 0.955 and 0.811 for normal eyes and 0.980 and 0.697 for eyes with RP, respectively [49]. Furthermore, the reliability and repeatability of split-detector AOSLO was 95.9% and 97.3%, respectively, for STGD, and 88.5% and 94.5%, for *RPGR*-associated RP, which was greater compared with confocal AOSLO [50]. In another study focused on cAOSLO-derived cone spacing, ICC for the interobserver agreement was 0.838 and 0.892 in normal and affected eyes, respectively. The ICC for inter-visit correlation was greater than 0.869 in both groups [51]. Cone density might show higher inter-individual and intra-individual eccentricity-dependent variability compared to spacing and hexagonality. Hence, a combination of all parameters is preferred over the use of a measure from the AO images [42].

#### 3.3.2. Blood Vessel

Structural evaluation of retinal vasculature can be performed using several metrics. The wall-to-lumen ratio (WLR) and cross-sectional area of the vascular wall (WCSA) are common metrics for structural changes in retinal blood vessels and are calculated using the outer and inner diameter of a blood vessel [52]. Using a rtx1 camera, Meixner et al. measured WLR, and it was independent of the total vessel diameter and depended mainly on the retinal vessel wall thickness [53]. Evidence shows that WLR is associated with age, hypertension, body mass index, and the stage of retinal microvascular abnormalities [53,54,55]. Meta-analysis on seven studies showed no significant difference in WCSA in hypertensive versus normotensive patients [56]. Microvascular density is another metric, which is calculated by dividing the total length of vessels in the ROI to the area of the region. Using AOSLO, lower microvascular density has been observed in diabetic macular ischemia and retinopathy [57]. Moreover, the bifurcation angle in the vascular-branching, tortuosity, which is calculated as total squared curvature divided by the length of the vessel, and the acircularity index of the FAZ as calculated by the perimeter of FAZ divided by the perimeter of the circle with equal area have also been described as metrics in AO imaging [4,5,6]. Functional evaluation is also feasible in measuring resting flow and stimulus-evoked changes in flow [7]. It is possible to create measures of vascular change and use them as biomarkers to disease progression and evaluate treatment response by tracking capillary perfusion and microaneurysms longitudinally [58].

## 4. AO Imaging in Diagnosing Early-Stage IRDs

Recently, the significant role of AO imaging has been highlighted in detecting the earliest changes at the cellular level in retinal diseases including dystrophies, many of which are detectable before clinical or pathologic findings of conventional retinal imaging modalities. Additionally, a better understanding of the natural course of retinal changes in dystrophic diseases can assist in the timely detection of patients who might be eligible for gene-specific intervention and in better selection of patients for clinical trials.

### 4.1. Rod-Dominated Dystrophies

Using confocal AOSLO in patients with RP, several authors reported a significant loss of cones in the central retina where the outer nuclear layer (ONL), ellipsoid zone (EZ), and fundus autofluorescence (FAF) are apparently normal [59,60]. In addition, cAOSLO of RP patients with preserved central vision showed only 1 out of 14 eyes with a normal cone mosaic, while 10 eyes had an irregular mosaic with large dark patches, indicating the disruption of the cone mosaic pattern. Cone density and the percent of the hexagonal Voronoi domain were significantly different from normal eyes. Reduced cone density was correlated with thinner ONL and photoreceptor layers [49,61]. Using the rtx1 camera, a significant decrease in foveal cone density has been reported in two patients with RP [61]. Comparing findings from OCT, FAF, and AOSLO in a patient with RP, three types of cone mosaic phenotypes were described: normal cone mosaic in the center of the fovea, blurred cone mosaic ring in the parafoveal area with rapid deterioration of the cone mosaic from the inner to the outer edge of the ring, and cone disappearance with visible RPEs. A dramatic decrease in cone density was observed while the density at the blurred ring area was impossible to determine [62]. Gale and colleagues reported AOFIO findings in 39 patients with RP using the rtx1 camera and described four stages of cone degeneration [63]. At wider eccentricities with a loss in the EZ band and thinned ONL on SDOCT, hypo-reflective blurred cones (stressed/dying cones) were noted on AO imaging. With increasing eccentricity, a mixture of hypo- and hyper-reflective spots with irregular spacing (photoreceptor cellular debris) was observed along with further thinning of the ONL. In areas of complete ONL loss, there were sparse hypo-reflective spots and no visible cones (RPE pigmentation) [63]. In a proof-of-concept study, Roshandel et al. showed that cone metrics are more sensitive than EZ span in detecting structural disease progression over 6 months in rod–cone dystrophy (RCD). Cone metrics should be considered as a potential structural endpoint for measuring disease progression and the efficacy of interventions in future RP clinical trials [13].

Early structural changes were observed in female carriers of the *RPGR* mutation using AO and microperimetry. Point-wise sensitivity at 68 loci located 1° to 9° away from the preferred retinal locus and cone density at 12 loci located 1° to 3° away from the foveal center were measured and showed severe point-wise sensitivity and cone density defects in asymptomatic patients with normal visual acuity [64]. In a study of 26 individuals with IRDs, cone density was decreased by up to 62% below normal at or near the fovea in eyes with normal visual acuity and foveal sensitivity. Despite cone density measurements being 52% below normal, visual acuity and foveal sensitivity were preserved and were negatively linked with cone spacing. Findings suggest that objective and direct measurements of the cone structure may be more sensitive predictors of disease severity than VA and foveal sensitivity [65].

In five patients with biallelic *Crumbs cell polarity complex component 1* (*CRB1*)-associated RP, loss of photoreceptors may be an early finding in asymptomatic patients [66], supporting that AO imaging may be more sensitive than conventional techniques, such as SD-OCT, in detecting subtle structural damage and diagnosing early-stage *CRB1*-associated retinopathy. Similarly, Sun et al. found a significant reduction in foveal cone density in patients with Usher syndrome compared with patients with RP in locations with normal EZ and the interdigitating zone (IZ) before visual acuity changes or OCT disruptions [67]. A cross-sectional study on eight family members with novel *Retinol Dehydrogenase 12* (*RDH12*) mutation showed reduced cone densities along the temporal meridian. Noticeably, one family member with a minimal clinical complaint and normal macular structure on SD-OCT showed a reduction in cone densities at all regions of interest temporal to the fovea on AOSLO [68]. Visual acuity and contrast sensitivity tests can be used to estimate the foveal cone function in patients with IRDs; however, contrast sensitivity appeared to be more sensitive than visual acuity in assessing the central visual function in patients with RP [69,70]. Strong correlations between visual acuity and cone spacing have been observed [36,71] Cone spacing was significantly increased in IRD patients and was correlated with visual acuity, foveal sensitivity, and multifocal electroretinogram (mfERG) findings [71,72]. Examples of the cone mosaic in a healthy eye and patients with different stages of RP are shown in Figure 6 and Figure 7.

### 4.2. Cone-Dominated Diseases

In the setting of mild disease in two adolescent relatives with autosomal dominant cone rod dystrophy with *Guanylate Cyclase Activator 1* (*GUCA1A*) mutation, AOSLO revealed cones with diminished reflectance relative to surrounding rods (described as “dark cones”) possibly due to the absence of or shorter outer segments [73,74]. In a cone–rod dystrophy (CRD) patient, AOFIO and AOSLO revealed large areas devoid of wave-guiding cones within the atrophic regions and abnormally large cones in a clinically spared location which were correlated with mfERG peak amplitudes [36]. In a non-syndromic biallelic *POC1 centriolar protein homolog B* (*POC1B)* mutation in a young male, a non-specific altered cone mosaic pattern with sparsely distributed cones around the central fovea was observed by AOFIO. Authors suggest that the disruption of the photoreceptor outer segments with preservation of the inner segments may be a feature of CD or CRD related to *POC1B* mutations, prompting evaluation with SD-AOSLO in CD patients [75]. In other studies, individuals with CRD showed increased cone space and decreased cone density [71,76,77]. Findings of AO in patients with enhanced S-cone syndrome include higher cone density at temporal parafovea with lower total photoreceptor densities, diminished outer segment cone receptors wave-guide signals, disrupted cone arrangement, larger cone cells with dark patchy-like lesions in the macula [78,79]. In some cases, AO imaging can be very helpful to distinguish specific photoreceptor diseases with closely shared features and almost the same clinical findings on standard evaluations. Moore et el. highlighted this beneficial role of AOSLO in differentiating *RGS9*/*R9AP*-associated retinal dysfunction (bradyopsia) (showing preserved integrity of cone mosaic at the fovea) with oligocone trichromacy (showing sparse cone mosaic at the fovea), both of which are similar in clinical and standard ERG findings [80].

### 4.3. Macular Dystrophies

#### 4.3.1. Stargardt Disease

AOSLO has elucidated early changes in Stargardt disease (STGD) which begin in photoreceptor outer segments spreading later to RPE cells even in locations with normal-appearing OCT and fundus autofluorescence [81,82]. The earliest cone spacing abnormality was observed in regions of homogeneous FAF, normal outer retinal structure with normal visual function. Closer to the fovea, cone spacing was increased with focal increased FAF near the central region of atrophy. AOSLO reflectivity did not correlate with the quantity of lipofuscin accumulation in RPE cells. In regions of complete macular atrophy, AOSLO showed bright oversaturated areas where cones were not seen, consistent with SD-OCT scans revealing extensive loss in the outer retinal layers [2]. In AOSLO imaging of a female with STGD, there was a lack of cone mosaic with visible RPE cells in the atrophic zone, blurred cone mosaic, and reduced visibility of photoreceptors with some areas of the starry-night cone pattern in increased FAF peri-atrophic areas and normal cone mosaic in normal FAF peri-atrophic area along with decreased cone density from the center to peripheral retina [62]. In a study on 14 cases of clinically diagnosed STGD, Razeen et al. suggested that compared to ONL, EZ, or IZ integrity assessment in OCT cone density measures by AO had a stronger correlation with retinal functional. They hypothesized that the significant cone loss in areas of intact EZ was attributed to the presence of abnormally enlarged rod photoreceptors [83].

#### 4.3.2. Vitelliform Macular Dystrophies (VMDs)

As shown by cAOSLO and SD-AOSLO imaging, cone photoreceptor inner segments are enlarged, and cone density is reduced within clinically apparent lesions. In contrast, both density and appearance of cone inner segments return to normal immediately adjacent to the vitelliform lesion. Split-detector imaging also revealed mobile disk-like structures, which may represent reactive subretinal cells [84]. Likewise, using multimodal AO within macular vitelliform lesions, both cone and RPE densities were reduced below normal. Outside of lesions, cone and RPE densities were slightly reduced but with a high degree of variability across the four genes *BEST1*, *PRPH2*, *IMPG1*, and *IMPG2* [85]. The degree of cone mosaic disruption in Best vitelliform macular dystrophy, although present in all stages, varied by the stage of the disease and was often patchy with areas of significant photoreceptor disruption surrounded by areas of a contiguous photoreceptor mosaic, even in a patient with advanced atrophy and fibrosis [86].

#### 4.3.3. X-Linked Retinoschisis (XLRS)

Using a multimodal AO retinal camera (MAORI, Physical Sciences, Inc., Andover, MA), offset aperture imaging showed “hill and valley” topography resembling the “spoke-wheel” pattern typically seen in XLRS. Significantly larger cones and increased cone diameter with greater eccentricity compared to healthy subjects were observed [87]. The authors suggest the larger cone diameters are due to trapped retinoschisin in the inner segment of the receptors. Duncan et al. explained the larger cones resulted from abnormal development of the fovea and chronic disruption due to intraretinal fluid accumulation. These authors argue that if the retinoschisin accumulation was the cause, the pathologic lesions would not be limited to the fovea region but extend to the whole retina [88]. Retinal AO-OCT images show mottling, a possible sign of abnormalities in the first synapse consistent with OPL dysfunction. Notably, even in clinically normal regions of the retina where the outer plexiform layer (OPL) seemed normal, AO-OCT indicated severe abnormalities at the OPL [87]. The width of the spoke-wheel retinal folds in the AO image was thinner than that in the fundus photographs [89]. Duncan et al. suggested that although cone density is reduced, the preservation of waveguiding cones at the fovea and eccentric macular regions has prognostic and therapeutic implications for XLRS patients [88].

### 4.4. Choroideremia

AOSLO imaging showed cone mosaic remains intact up to the border of retinal atrophy, which was abnormally heterogeneous in morphology, diameter, and density sharply at the border [90,91], as well as sharp functional transitions between healthy and degenerated retina found by AO microperimetry [92]. Subclinical widespread large RPEs with increased inter-center distance between the cells were observed. The presence of patches of intact cone photoreceptors on AO-enhanced indocyanine green (AOICG) indicate the disruption of the RPE blood–retinal barrier function [93]. Correlation between cone spacing and choriocapillaris flow voids, and negative correlation with retinal sensitivity, may suggest cone degeneration accompanied by reduced choriocapillaris perfusion [94]. Other distinct features of choroideremia were lower image contrast, less well-defined cone edges, and groups of cones with high reflectance relative to the surrounding cone mosaic, compared to healthy eyes [90]. Bubble-like lesions as first described by Morgan et al. appeared as hyper-reflective spots with dark edges, which co-locate with hypo-reflective spots in the choroid in OCT [90,91]. Large, sparse, and remnant cone inner segments within outer retinal tubulations and the continuity of the ORTs with the preserved retina may suggest an intermediate stage of retinal degeneration in choroideremia [91]. Recent advancements in multimodal visualization of human RPEs are promising for *CHM* gene therapies [95,96,97]. Table 2 summarizes the findings of AO in IRDs.

## 5. AO imaging in Tracking the Progression of IRDs

A longitudinal study on three patients with IRDs (two RP and one Usher) showed a significant decline in cone density (9.1% per year) in sham-treated eyes compared to eyes treated with the ciliary neurotrophic factor (CNTF). Similar changes were observed in OCT but not visual acuity, visual field, and ERG [3]. Cone metrics along with microperimetry were useful for assessing structure–function correlations in normal subjects and carriers of X-linked RP, at a cellular level [98]. In a study on *RP1*-associated autosomal recessive RP patients, Ueno et al. reported that despite the lack of cone mosaic visibility, homogenous residual islands of photoreceptors, in accordance with hyper-autofluorescent areas in FAF, were seen in AO fundus images with small hypo-reflective clumps at the edges. The size and homogeneity of these areas were diminished in follow-up images over 4 years [99]. In a 2.5-year follow-up of two patients with Best vitelliform macular dystrophy in regions directly around lesions, photoreceptor density was normal and steady [84]. In a longitudinal study over 21 to 29 months on 3 cases of *RPE65*-associated Leber congenital amaurosis (LCA), Kalitzeos et al. found that while there were no visualized changes in confocal AOSLO, non-confocal split-detection showed inner segment enlargement along with a reduction in foveal cone density over time [100]. In AOSLO imaging of an STGD patient taken over 1 year, enlarging outer retinal tabulations—a suggestive sign for a more advanced stage of the disease—were seen as finger-like projections with central teardrop structures in split-detection but not confocal AO imaging. As the length of tubulation increased, the central teardrop-like structures, which were assumed to constitute migratory RPE cells, seemed to reduce in size, in line with morphologic changes in spindle-shaped RPE cells in later stages of degenerative changes [83]. AO has also been used for tracking ultra-rare IRDs, such as *RCBTB1*-associated retinopathy showing progressive irregularity or complete loss of cone mosaic in areas of RPE and outer retinal atrophy [101]. In a study on patients with resolved central serous chorioretinopathy, cone density was remarkably decreased compared to normal eyes, but it significantly increased 12 months after resolution [102]. In another study, authors reported the regeneration of cone cells 12 months after successful retinal detachment repair (scleral buckling) using a rtx1 AO camera [103].

## 6. Limitations

### 6.1. Resolution

As previously mentioned, due to the relatively lower transverse resolution of AOFIO compared with AOSLO, small cones of central 2° and rods are not reliably detectable by this technology (Figure 8B). Such a low axial resolution can decrease image contrast as well as the repeatability of cone density measurements in longitudinal assessments [104]. Although AOSLO cameras can resolve rods and foveal cones in a very small area of the retina, the lengthy process of multiple frame acquisition in the macular region and the extensive montage required make this technology less practical in the clinical setting. It is also notable that regardless of image quality most rods and some foveal cones cannot be visualized in SD-AOSLO due to contrast issues affecting image resolution and the photoreceptor refractive index profile [23]. Speckled noise in OCT imaging can also lead to a diminished resolution in AO-OCT, most of which can be improved using a variety of post-processing methods of image deblurring or more recent deep learning-based methods [105].

### 6.2. Image Quality

AO Image quality is affected by a variety of instrument-related factors, including acquisition speed and coherence properties of the illumination source as well as the patient’s ocular conditions, including refractive errors, corneal scars, cataract or prior intraocular lens implantation, rapid eye movements, age-related miosis, the severity of retinal diseases, and the presence of coexisting retinal abnormalities (Figure 8C–F). Moreover, it has been revealed that image quality decreases with an increasing retinal eccentricity [106]. By contrast, many imaged cones may not be sharp enough to be easily detectable because of their altered directionality and diminished waveguiding properties despite intact structures [107]. Distortions and artifacts caused by eye motions are more problematic in higher magnifications and *en face* planes. Though some of these distortions are correctable in post-imaging processing, instruments with higher acquisition speeds that are less vulnerable to motion artifacts are preferred. Recently introduced AO-OCTs with 1 MHz A-scan have this qualification, but sacrificing image sensitivity at such a high speed makes a different challenge, especially for the visualization of inner retinal layers. So, the variety of IRD patients who can effectively be imaged by AO is limited, especially in the presence of concomitant ocular pathologies. Backscattering of light from cellular debris or other outer retinal cells, such as RPEs, increases the signal-to-noise ratio and reduces the image quality of a diseased area in these patients. To highlight the significance of limitations caused by poor AO image quality, it is worth noting that in numerous research studies on retinal diseases or even normal healthy retina, almost half of the taken AO images are reported to be excluded due to image quality issues [104,108,109]. Grading image quality and differentiation of low-quality images due to optical reasons from pathology-related mosaic disruption is a major challenge based on our observations [106]. Although an experienced observer can differentiate these two entities in most instances, mild optical distortions may not be detected and can be attributed to retinal pathology [13]. Hence, cautious interpretation of cone metrics should be considered in the presence of mild optical barriers, such as cataract, astigmatism, or inner retinal pathologies, which do not contraindicate with taking images of fairly good quality but might affect the results.

### 6.3. Localization

Accurate localization of the region of interest (ROI) is critical in retinal image analysis. Enhanced magnification of AO imaging results in a very small field of view, which creates a challenge for the precise localization of the ROI. Since the AO fixation point may not coincide with other retinal center landmarks that are more reliable, relying on the distance from the AO fixation point may cause errors in ROI localization. Our recent study showed a significant shift in the AO montage center in relation to the preferred retinal locus and foveal pit center, which could result in cone mosaic measurement errors [110]. Retinal vasculature is the most used landmark for aligning the AO montage with standard *en face* retinal images, such as near-infrared reflectance (IR) or fundus autofluorescence images. The IR + OCT module can be very useful in precise localization of the foveal pit center, which then can be marked on the IR image and used as a reference for AO alignment. Location of the preferred retinal locus can be determined by aligning the AO montage with the SLO image provided by microperimeters, such as MAIA; although, the SLO images might be distorted, thus preventing accurate alignment [110].

Other authors have attempted to overcome this limitation by conjugating a wide-field line-scanning ophthalmoscope and a closed-loop optical retinal tracker with AOSLO [111]. By further efforts toward implementing stabilization systems with digital registration [112] in optical eye tracking and hybrid tracking, AOSLO has provided real-time stabilization with the ability to correct high-amplitude low-frequency drifts of the eye in a wider field of view [113]. The importance of accurate localization and sequential image alignment is highlighted in follow-up visits and longitudinal assessments for patient tracking. Using modules with registration strategies, such as OCTs with intensity-based volumetric registration, spherical mapping of the surface, fovea alignment, or even point-to-point correspondence by in-exact matching of surface currents and landmarks on *en face* OCT, can be very helpful in this regard [114,115].

### 6.4. Structure–Function Correlation

To take full advantage of the AO imaging technology in the diagnosis and monitoring of IRDs, one has to consider incorporation and correlation of functional mapping with a photoreceptor mosaic. Prior studies have confirmed the strong structure–function correlation of photoreceptors in retinal diseases and specific IRDs [116,117] while others have shown dissociation of function and structure at the fovea in healthy retinae [98] and retinal pathologies [66,118,119]. Correlating photoreceptor mosaic properties with localized retinal function might be hindered by challenges in accurate co-localization of the ROIs used for the two measurements [64,110] and a lack of consistent relationship between photoreceptor measurements and retinal sensitivity across the macula [98]. The latter might be explained by the fact that in normal retina, the retinal light sensitivity is determined by the size of the receptive field of the retinal ganglion cells rather than the density or spacing of the photoreceptors [120]. However, a relationship between retinal sensitivity loss and photoreceptor degeneration has been reported in different retinal pathologies [116,117,121]. AO microperimetry (AOMP) is a new technology that incorporates AOSLO and microperimetry to map photoreceptor mosaic and retinal sensitivity simultaneously. AOMP can be very useful in the assessment of structure–function correlation at a cellular level in healthy and diseased retinae [92,122,123,124].

### 6.5. Cone Matching and Follow-Up Imaging

Short-term and long-term variations in cone reflectivity and dysflective cones have been noted in normal subjects with no known ocular disease [125,126]. It has been proposed that the changes in the reflectivity of cones are either the result of altered molecular and biological processes, such as phototransduction, or changes in outer segment length, which acts as a biological interferometer [127,128]. In addition, findings of our prior research have shown that overlapping images that cover the region of interest but were taken with different fixation coordinates may vary significantly in terms of image quality and cone metrics [13]. Although current technology enables follow-up examinations on the same ROI, the accuracy of automated ROI detection might be affected by a shift in the fixation point and/or change in retinal vascular pattern due to pathology. Hence, we recommend alignment of the AO montage with *en face* retinal scan to localize the ROI precisely.

### 6.6. Special Considerations in Patients with RP

Photoreceptor mosaic parameters are gaining popularity as structural endpoints in RP clinical trials, though improvements in imaging and image analysis protocols are required. The most important challenge facing the use of AO imaging in patients with RP is the frequent ocular co-morbidities, such as cataract (or history of cataract surgery), macular edema, high refractive errors, outer retinal atrophy, and epiretinal membrane and fixation instability that reduce the image quality and affect mosaic visualization (Figure 8C–F). Feasibility of AO imaging may vary with the properties of the patient cohort. For example, we reported that 80% of patients met one or more exclusion criteria and were not eligible for AO imaging [13]. However, other studies reported higher feasibility rates [63]. In addition, the development of new or the progression of an existing lens opacity or cystoid macular edema during the follow-up period are likely, which can reduce image quality and alter mosaic properties artifactually. The differentiation of actual photoreceptor mosaic changes due to disease progression from diminished image quality can be very challenging in these cases.

## 7. Conclusions

Overall, AO imaging remains a viable and potentially useful tool for early diagnosis and monitoring disease progression in IRDs. Using cone metrics as structural endpoints may help with an earlier detection of the efficacy of neuroprotective interventions in halting disease progression. More interestingly, AO imaging was useful in detecting disease progression in patients with RP and no obvious outer retinal defect on SD-OCT. Considering that in a group of IRDs, including RP, clinically significant visual loss tends to occur after foveal involvement, the early findings of AO in the foveal region can provide an early signal for the safety and efficacy of interventions for IRDs. Upon further validation, AO imaging has the potential to revolutionize structural outcome measures used in future clinical trials on RP and other IRDs. Lack of standard image acquisition and analysis protocols is one of the main issues that should be addressed in future studies using AO retinal cameras. While imaging of the temporal retina along the horizontal meridian may be a good option for monitoring disease progression in RP, central images can be more useful in detecting parafoveal photoreceptor mosaic changes at early stages of macular disorders or generalized retinopathies.

## Figures and Tables

**Figure 1 diagnostics-13-02413-f001:**
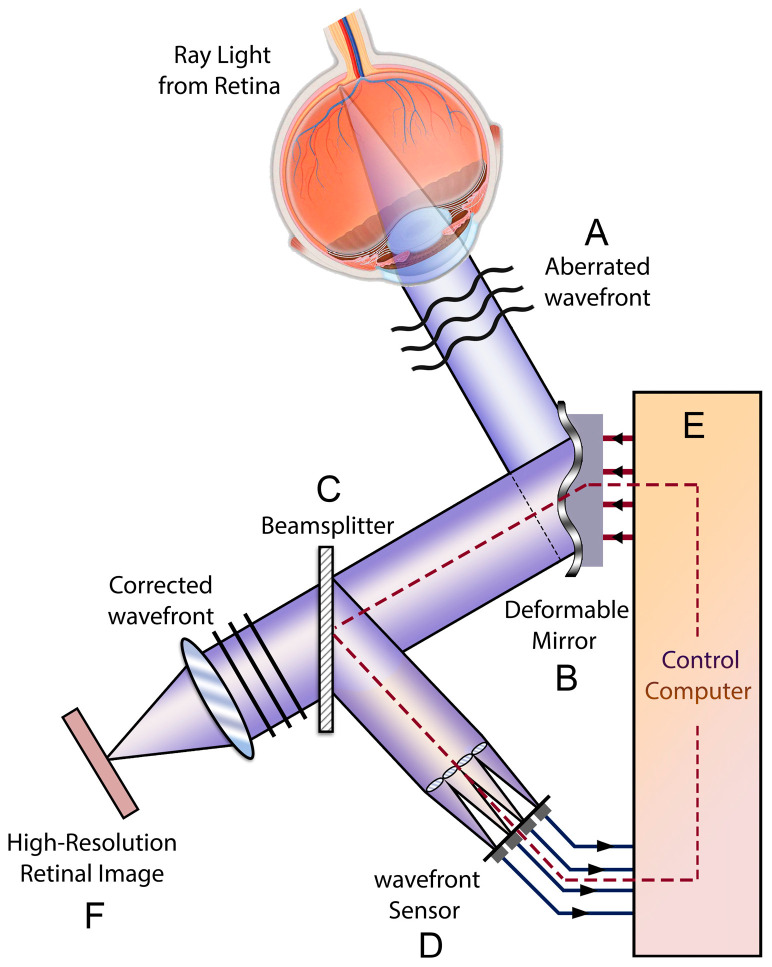
Schematic view of the adaptive optics retinal camera. The aberrated wavefront exits the eye (**A**), and the aberrations are corrected by a deformable mirror (**B**). The resultant image is divided by a beam splitter (**C**) and equally received by a wavefront sensor (**D**) measures the residual aberrations of the image corrected by the deformable mirror. Aberrometry data are analyzed by computer software (**E**), which adjusts the deformable mirror, and this loop (B > C > D > E > B) continues working until the least amount of aberration is detected by the wavefront sensor, when the final image will be captured and recorded by the retinal camera (**F**).

**Figure 2 diagnostics-13-02413-f002:**
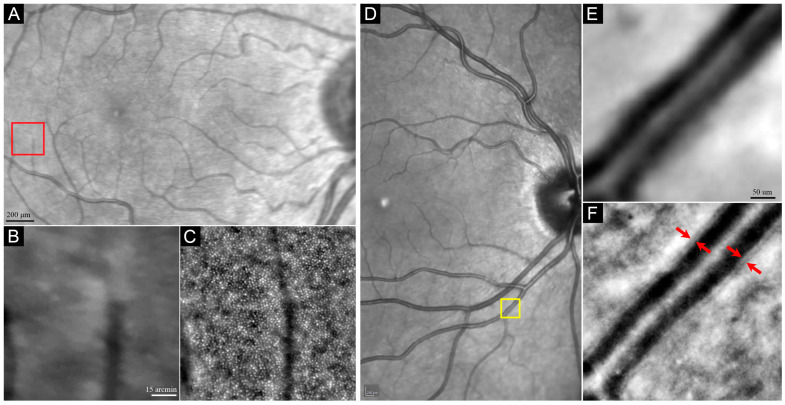
Comparison of the magnified infrared reflectance (IR) image taken by SLO camera (**A**,**B**,**D**,**E**) with AO image taken by rtx1 AO camera (**C**,**F**). Red arrows in F show the internal and external boundaries of the vessels wall. Red (**A**) and yellow (**D**) boxes represent the magnified area shown in panels (**B**,**C**) and (**E**,**F**), respectively.

**Figure 3 diagnostics-13-02413-f003:**
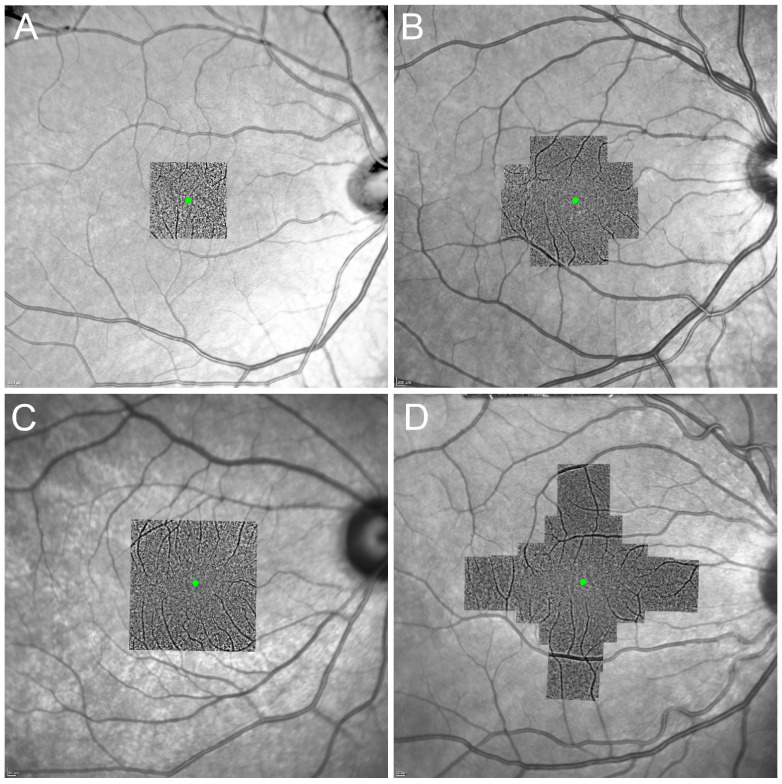
Examples of montage created from different AO image acquisition sequences. Overlapping images were taken to cover the central 3° from the fovea using 4 image acquisitions (**A**), the central 6° using 12 (**B**) or 16 (**C**) image acquisitions, and the extended imaging along the horizontal and vertical meridians (up to 9°) using 20 image acquisitions (**D**). Green dots represent the foveal center.

**Figure 4 diagnostics-13-02413-f004:**
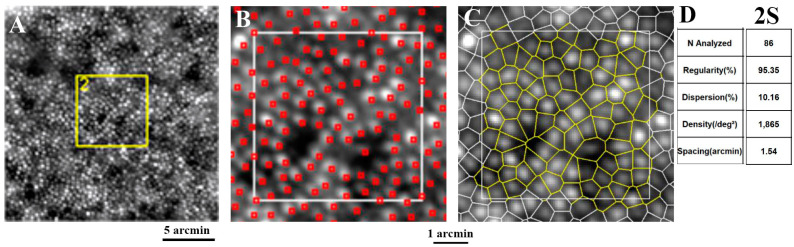
AO imaging using rtx1 device (**A**). The region of interest (yellow square in panel (**A**)) was used for automated cone detection (**B**) and segmentation (**C**) by AODetect software. Cone mosaic parameters are shown in (**D**). The analysis was performed at superior 2° (**2S**) from the fovea. White squares in panels (**B**,**C**) represent the analysis area.

**Figure 5 diagnostics-13-02413-f005:**
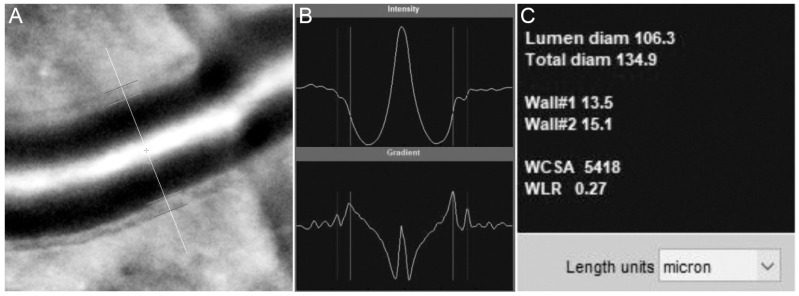
The internal and external vessel wall boundaries are marked automatically by the AODetect software (**A**) and adjusted manually if required. The boundaries are determined based on the peak signals (**B**) and measurements, including lumen diameter, total diameter, wall thicknesses, wall cross-sectional area (WCSA), and wall-to-lumen ration (WLR), which are provided in microns (**C**).

**Figure 6 diagnostics-13-02413-f006:**
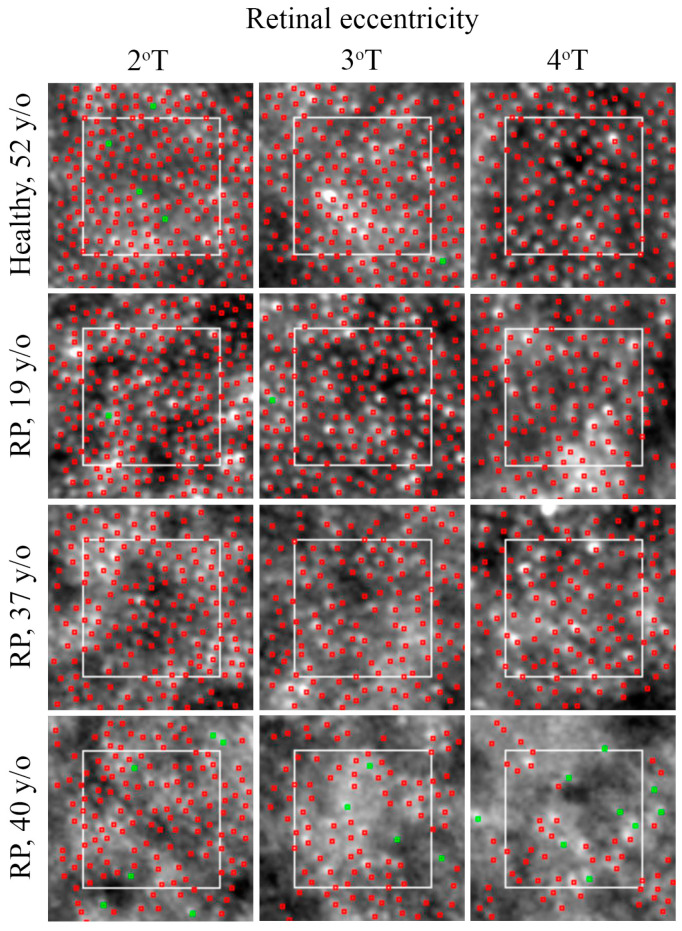
Cone mosaic changes in RP. Automated detection of cones (red dots) with manual adjustment (green dots) shows normal cone mosaic with an eccentricity-dependent decline in cone density and an increase in cone spacing (**top row**) in a healthy subject. Patients may reveal mild perifoveal mosaic alteration (**second row**), parafoveal and perifoveal alteration (**third row**), and severe parafoveal and perifoveal alteration (**bottom row**). T = temporal; RP = retinitis pigmentosa.

**Figure 7 diagnostics-13-02413-f007:**
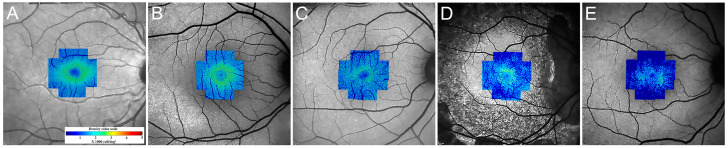
Cone density map in a 28 y/o healthy control (**A**) and different stages of RP (**B**–**E**). Note the increased cone visibility at the fovea in patients with RP (most prominent in **B**), a phenomenon that has been reported in these patients.

**Figure 8 diagnostics-13-02413-f008:**
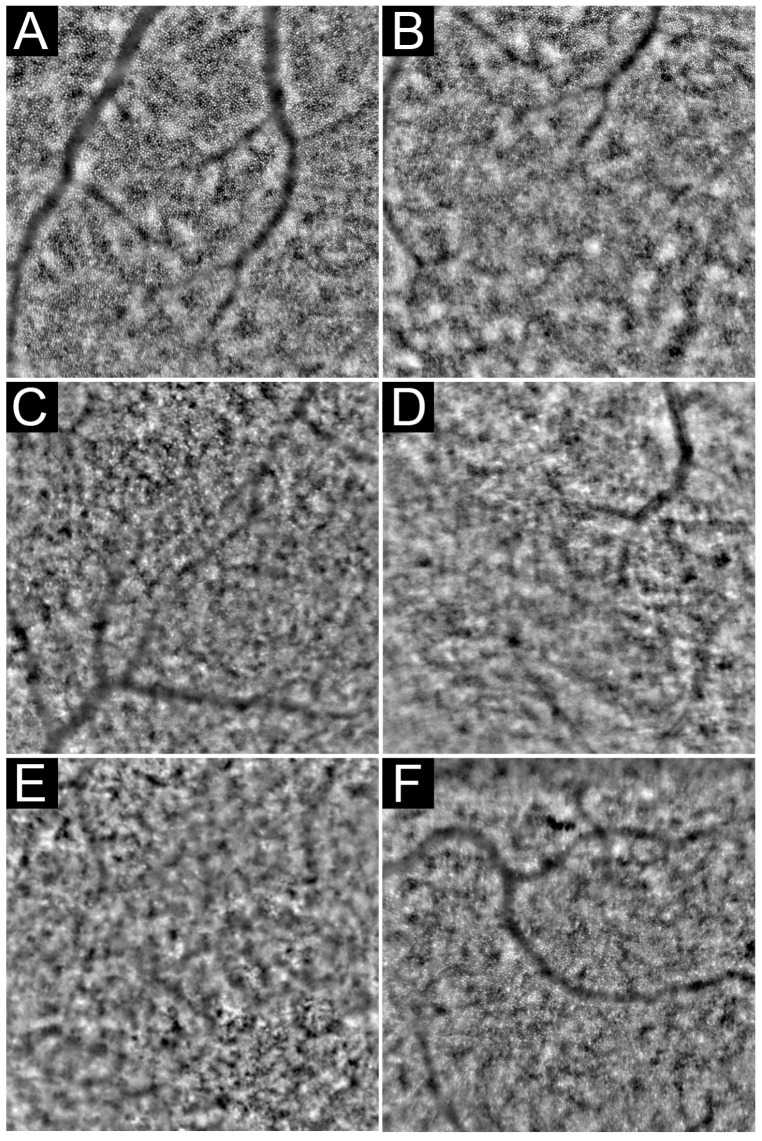
Examples of AOFIO. Normal cone mosaic detectable in the entire imaging field in a normal healthy retina (**A**). Foveal cones are undetectable within the central 1–2° (**B**). Image quality might be affected by visually significant cataract (**C**,**D**), cystoid macular oedema (**E**), or high astigmatism (**F**).

**Table 1 diagnostics-13-02413-t001:** Comparison of Flood Illumination and Scanning Laser Ophthalmoscope.

	FIO	SLO
Availability	Commercial	Research custom-made
Field	Large (4°)	Small (0.2–1.0°)
Illumination	Diffuse	Focused single spot
Imaging technique	Single flash	Scanning laser
Transverse resolution	2–3 µm	1–2 µm
Axial resolution	200–300 µm	100 µm
Motion artefacts	Low	high
Visible structures		
Cones	Yes ^a^	Yes
Rods	No	Yes
RPE	Yes ^b^	Yes
RGC	No ^c^	Yes
Retinal vessels	Yes	Yes
Choriocapillaris	No ^c^	Yes

^a^ Unable to resolve foveal cones in the central 1–2°. ^b^ Only by transscleral illumination. ^c^ Might be visualized using multiply scattered light [23]. FIO: flood illumination ophthalmoscopy, SLO: scanning laser ophthalmoscopy, RGC: retinal ganglion cells, RNFL: retinal nerve fiber layer.

**Table 2 diagnostics-13-02413-t002:** Summary of AO imaging findings in IRDs.

IRD Type	AO Technology	Findings of AO Imaging
Retinitis pigmentosa	AOSLO, AOFIO	Significant cone loss in the central retina despite intact ONL, EZ, and normal FAF, irregular cone mosaic, lower cone density, decreases the percent of hexagonality, hypo-reflective blurred cones in eccentricities, increased cone spacing, visible RPE, and no detectable cone mosaic in areas of complete INL loss
*RPGR* Female carrier	AOFIO	Irregular cone mosaic and decreased cone density in asymptomatic patients with normal visual acuity
*CRB1*	AOFIO	Decreased cone density despite normal visual acuity and OCT
*RDH12*	AOSLO	Reduced cone density along the temporal meridian
*GUCA1A*	AOSLO	Truncated cone in outer segments, reduced cone reflectance (dark cones)
*POC1B*	AOFIO	Disruption of photoreceptor in outer segments with preservation of the inner segments blurred EZ and significant cone mosaic disruption around the fovea
*NR2E3* (Enhanced S-cone syndrome)	AOSLO	Higher cone density at temporal parafovea with lower total photoreceptor densities, diminished outer segment wave-guide signals, disrupted cone arrangement, larger cone cells with dark patchy-like lesions in the macula
*RGS9/R9AP* (bradyopsia)	AOSLO	Reserved integrity of cone mosaic and cone density within the normal range. A small area of central hypo-reflective lesion of non-waveguiding cones with minimal decrease in peak cone density reported in a patient
*ABCA4* (Stargardt disease)	AOSLO	Disrupted cone mosaic, decreased cone density from central to peripheral retina, increased cone spacing in regions with normal OCT and FAF, starry-night cone pattern and increased cone spacing in areas of increased FAF, lack of cone mosaic with visible RPE cells in the atrophic zone, significant cone loss and abnormally enlarged rod photoreceptors despite intact EZ, no correlation between the quantity of lipofuscin accumulation in RPE cells and reflectivity shown in AOSLO
VMD	AOSLO	Reduced cone and RPE density and enlarged inner segments within the vitelliform lesion, reactive subretinal cells shown as mobile disk-like structures, patchy areas of significant photoreceptor disruption surrounded by contiguous photoreceptor mosaic
*RS1* (XLRS)	AOSLO, AO-OCT	Spoke-wheel appearance in offset aperture imaging, increased cone diameter, decreased cone density, severe abnormalities at the OPL in AO-OCT of clinically normal regions, no detectable cone mosaic in areas of EZ and IZ loss
*CHM* (Choroideremia)	AOSLO, AOICG, AO-OCT	Intact cone mosaic up to the border of retinal atrophy, ill-defined cone edges, groups of cones with high reflectance relative to the surrounding cone mosaic, bubble-like hyper-reflective spots with dark edges, large remnants of cone inner segments within outer retinal tubulations, widespread large RPEs, choriocapillaris flow voids in areas of increased cone spacing, disruption of RPE blood barrier in AOICG

AO: adaptive optics, AOFIO: adaptive optics flood illumination ophthalmoscopy, AO-OCT: adaptive optics coherence tomography, AOICG: AO-enhanced indocyanine green, AOSLO: adaptive optics scanning laser ophthalmoscopy, EZ: ellipsoid zone, FAF: fundus autofluorescence, ICG: indocyanine green, INL: internuclear layer, IZ: interdigitation zone, mfERG: multifocal electroretinogram, OCT: optical coherence tomography, OPL: outer plexiform layer, ONL: outer nuclear layer, RPE: retinal pigment epithelium; VMD = vitelliform macular dystrophy; XLRS = X-linked retinoschisis.

## Data Availability

Not applicable.

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
