# Peer review of "Pearls and Pitfalls of Adaptive Optics Ophthalmoscopy in Inherited Retinal Diseases"

_diagnostics, 2023, doi:10.3390/diagnostics13142413_

Round 1

Reviewer 1 Report

In this manuscript, Ashourizadeh et al. reviewed current AO techniques and their applications for retinal imaging. AOFIO, AOSLO, and AOOCT are the main technical focuses of the manuscript, and their clinical and research imaging results are presented. Most importantly, related diseases such as rod-dominated dystrophies, cone-dominated diseases and choroideremia, are intensively reviewed. Finally, current limitations of AO techniques are summarized. In addition to the current content, in the conclusion and discussion part, I wish the authors could give more insights on how to build standardized imaging protocols while applying AO to clinical cases.

Additional minor comments are:

·       Line 190-191, and line 201-202, please specify possible purposes of photoreceptor imaging, and their corresponding imaged area. Some examples might be helpful for understanding.

·       Line 302, please give the full name of FAZ before using it. Please check other abbreviations as well.

Minor editing of English language required. Please check and make sure full names are given before first use of the abbreviations.

Reviewer 2 Report

This article has a good practical base and  motivation  :incorporation of the adaptive optics (AO) technology into ophthalmic devices has led to the development of non-invasive ultra-high-resolution imaging of retinal cells and blood vessels with minimal image degradation by higher-order monochromatic aberrations. It s well structureted into  basic concepts consist into  principles of adaptive optics,technologies and devices; acquisition, analysis, and interpretation and a separated chapter  of AO imaging in diagnosing early-stage IRDs and AO imaging in tracking the progression of IRDs.

The authors have a real procupation to be an   organized and comprehensively described article with appropriate and adequate references.

This article is an important page of theoretical review with a practical implication: AO imaging remains a viable and potentially useful tool for early diagnosis and monitoring disease progression in IRDs. Using cone metrics as structural endpoints may help with an earlier detection of the efficacy of neuroprotective interventions in halting disease progression.

All of these arguments make that the article been been  accept in present form.

Author Response

We thank the reviewer for their positive and inspiring feedback.

Reviewer 3 Report

Dear authors:

Thanks for your working hard and therefor, we could get know more.

Best regards 

                  Chi-Ting Horng  MD, PhD   

Author Response

We thank the reviewer for their positive feedback.